# Detection of a DNA Methylation Signature for the Intellectual Developmental Disorder, X-Linked, Syndromic, Armfield Type

**DOI:** 10.3390/ijms22031111

**Published:** 2021-01-23

**Authors:** Sadegheh Haghshenas, Michael A. Levy, Jennifer Kerkhof, Erfan Aref-Eshghi, Haley McConkey, Tugce Balci, Victoria Mok Siu, Cindy D. Skinner, Roger E. Stevenson, Bekim Sadikovic, Charles Schwartz

**Affiliations:** 1Department of Pathology and Laboratory Medicine, Western University, London, ON N6A 3K7, Canada; shaghsh@uwo.ca; 2Molecular Genetics Laboratory, Molecular Diagnostics Division, London Health Sciences Centre, London, ON N6A 5W9, Canada; michael.levy@lhsc.on.ca (M.A.L.); jennifer.kerkhof@lhsc.on.ca (J.K.); haley.mcconkey@lhsc.on.ca (H.M.); 3Children’s Hospital of Philadelphia, Philadelphia, PA 19104, USA; arefeshghe@chop.edu; 4Department of Paediatrics, Western University, London, ON N6A 3K7, Canada; tugce.balci@lhsc.on.ca; 5Medical Genetics Program of Southwestern Ontario, London Health Sciences Centre, London, ON N6A 5W9, Canada; victoria.siu@lhsc.on.ca; 6Greenwood Genetic Center, Greenwood, SC 29646, USA; cskinner@ggc.org (C.D.S.); res@ggc.org (R.E.S.)

**Keywords:** epigenetics, DNA methylation, episignature, *FAM50A*, Armfield X-linked intellectual disability, constitutional disorders

## Abstract

A growing number of genetic neurodevelopmental disorders are known to be associated with unique genomic DNA methylation patterns, called episignatures, which are detectable in peripheral blood. The intellectual developmental disorder, X-linked, syndromic, Armfield type (MRXSA) is caused by missense variants in *FAM50A*. Functional studies revealed the pathogenesis to be a spliceosomopathy that is characterized by atypical mRNA processing during development. In this study, we assessed the peripheral blood specimens in a cohort of individuals with MRXSA and detected a unique and highly specific DNA methylation episignature associated with this disorder. We used this episignature to construct a support vector machine model capable of sensitive and specific identification of individuals with pathogenic variants in *FAM50A*. This study contributes to the expanding number of genetic neurodevelopmental disorders with defined DNA methylation episignatures, provides an additional understanding of the associated molecular mechanisms, and further enhances our ability to diagnose patients with rare disorders.

## 1. Introduction

Mendelian neurodevelopmental disorders usually present with developmental delay (DD), intellectual disability (ID), and/or congenital anomalies (CA). These syndromes are often associated with complex and overlapping symptoms including overgrowth, aberrant craniofacial features, seizure, and neurological abnormalities, which may complicate clinical diagnosis [1]. The frequency of Mendelian disorders is approximated to be 40 to 82 per 1000 live births [2]. Considering all congenital anomalies, 8% of individuals are estimated to have a genetic disorder before adulthood [3]. Given the broad range of genetic and phenotypic heterogeneity, based on an individual’s presentation and clinical assessment alone, it is often impossible to determine the precise clinical diagnosis in the absence of a specific molecular genetic diagnosis. Conventional genetic testing, including the analysis of sequence and copy number variants (CNVs) and comprehensive genome-wide methods such as whole exome sequencing (WES), leaves a substantial proportion of subjects unresolved [4]. Genetic analysis in patients with a confirmed clinical diagnosis often yields no significant genetic findings or results in genetic variants of unknown clinical significance (VUS). 

Epigenetics is defined as the study of heritable alterations in DNA that do not involve the DNA sequence. Recent advances in epigenetic analysis have provided an alternate approach for diagnosis of genetic disorders. Pathogenic variants in genes that encode proteins involved in the epigenetic machinery, chromatin assembly and transcription regulation can cause changes in the genome-wide pattern of DNA methylation, differentiating them from unaffected individuals [5]. These highly sensitive and specific changes in DNA methylation patterns, referred to as episignatures, are currently used to help reclassify VUS’s as likely pathogenic or benign, thus enabling a definitive diagnosis [6]. Hence, a term episignature is used to describe a consequence of a unique DNA methylation pattern, resulting from the underlaying DNA mutation.

Episignatures have the potential to provide insights into the functional effects of DNA methylation variation and its association with pathophysiology of a disorder. We and others have demonstrated the utility of episignatures in diagnosing rare genetic disorders. More than 30 different genetic syndromes associated with mutations in over 50 genes have been described that exhibit specific DNA methylation episignatures [7,8,9,10,11,12,13,14,15,16]. These episignatures can overlap. For instance, CpG sites located in *HOXA5* were reported to be similarly hypermethylated in CHARGE and Kabuki syndromes [9]. At one extreme, DNA methylation episignatures can be identical across multiple genes belonging to common protein complexes. As an example, Coffin–Siris syndromes (CSS), Nicolaides–Baraitser syndrome (NCBRS), and Chr6q25 microdeletion syndrome, commonly known as BAFopathies, which arise from SWI/SNF remodeling complex defects, share a common, highly overlapping episignature. Interestingly, the overlap between the methylation pattern of some subtypes of CSS and NCBRS are higher than the overlaps found within some CSS subtypes [17]. At the other extreme, we have identified multiple distinct episignatures in single genes. Patients with ADNP syndrome exhibit two distinct DNA methylation profiles associated with two separate protein domains [18]. These genome wide episignatures are the consequence of genetic mutations resulting in a defective function of the related protein. Regions with significant disruptions in DNA methylation can range from hundreds to tens of thousands of probes in the methylation array, but can show partial overlap between different disorders, and normally do not involve disruption of DNA methylation in the related gene [19]. 

The intellectual developmental disorder, X-linked, syndromic, Armfield type (MRXSA) is a rare genetic disorder, described first in 1999 [20]. Patients inherit this syndrome in an X-linked recessive fashion. Affected males manifest symptoms including intellectual, skeletal, ocular, and craniofacial abnormalities, while carrier females have no clinical symptoms. In infancy to early childhood, patients represent seizures [20]. All individuals exhibit a degree of global developmental delay, presenting with impaired speech, difficulty in walking, and/or a need for special education. The skeletal abnormalities include short stature, small hands and/or feet, joint hypermobility, stiff joints, and/or club foot. Most patients have ocular anomalies, such as glaucoma, strabismus, nystagmus, exotropia, and/or keratoconus. The craniofacial abnormalities include macrocephaly, epicanthal folds, depressed nasal bridge, downslanted palpebral fissures, cleft palate, bow-shaped mouth, microretrognathia, broad forehead, micrognathia, infraorbital creases, wide nasal root, short and lightly upturned nose with underdeveloped nares, posteriorly rotated ears, faint hemangiomas between brows and at back of neck, bulbous nose, prominent tall forehead, and/or overfolded helices [20,21]. Armfield et al. attributed the condition to an 8 Mb region on Xq28, using linkage analysis [20]. Recently, rare hypomorphic missense variants in *FAM50A* (family with sequence similarity 50 member A) have been identified as the causal variants for the disorder in this region (Xq28) [21]. Defects in *FAM50A* are established to cause aberrant spliceosome C complex function, defining MRXSA as a spliceosomopathy [21]. 

In this study, we performed genome-wide DNA methylation analysis to assess if an episignature was associated with MRXSA. By comparing the methylation data of patients with matched normal controls, a specific DNA methylation profile was identified. Using these data, we developed a support vector machine (SVM) classifier for this disorder. This classifier enables the identification of individuals with likely pathogenic variants in *FAM50A*. We also demonstrated the high specificity of the *FAM50A* episignature by comparing it to over 1000 samples from patients with episignatures in over 40 genes associated with 38 other neurodevelopmental syndromes.

## 2. Materials and Methods 

### 2.1. Subjects and Cohorts

DNA samples were extracted from peripheral blood of six individuals from three different families with a confirmed diagnosis of MRXSA, all recruited from the Greenwood Genetic Center (Greenwood, SC, USA). For one of the patients, two samples were available, extracted at different ages. The newer sample was used for the purpose of selecting the significant probes and training the classification model, and the older sample was used as a technical control sample. All the samples and records were de-identified. The research was conducted in accordance with the Declaration of Helsinki. The study protocol was approved by the Western University Research Ethics Board (REB 106302; REB 116108). Physicians obtained informed consent from the aforementioned patients for use of the clinical information.

### 2.2. Methylation Data Analysis

We performed DNA methylation analysis of the samples after bisulfite conversion, using Illumina Infinium methylation EPIC bead chip arrays, according to the manufacturer’s protocol. These arrays include over 850,000 CpG sites in the human genome. Details of the methylation data analysis are previously described [4,8,19]. In summary, intensity data files comprising the methylated and unmethylated signal intensities were analyzed in R 4.0.2. We normalized the methylation data based on the Illumina normalization method with background correction using the minfi package [22]. We eliminated the following probes: Detection *p*-value > 0.01; X and Y chromosome probes; contained single nucleotide polymorphisms (SNPs) at or near CpG interrogation sites; or cross-reactive with other genomic regions. The removal of these probes is performed in order to ensure that the difference observed between the case and control groups is only due to DNA methylation changes rather than other factors. Using the MatchIt R package [23], for each case we selected seven controls matched for age, sex, and array type from the EpiSign Knowledge Database (EKD) [19]. The number of control samples was increased until the matching quality reached an optimum point, and the ratio of seven to one proved to be the most appropriate choice. Principal component analysis (PCA) was used to check for outlier case and control subjects and examine the batch effect. 

### 2.3. Probe Selection, Dimension Reduction, and Constructing a Supervised Classifier

Methylation levels calculated as the ratio of methylated signal intensity over the sum of methylated and unmethylated signal intensities, called the β-values, were converted to M-values by logit transformation using the formula log_2_(β/(1-β)) to obtain homoscedasticity for linear regression modeling using the limma package [24]. The model matrix was constructed by these values. We added as confounding variables the estimated blood cell proportions derived by the algorithm developed by Houseman et al. [25]. Next, eBayes function was operated in order to moderate the generated *p*-values. We performed the probe selection process in three steps. First, we selected 1000 probes with the highest product of methylation differences between case and control samples and the negative of the logarithm of multiple-testing corrected *p*-values derived from the linear modeling by Benjamini–Hochberg (BH) method. The advantage of this approach over setting strict cut-off values for the *p*-value and methylation difference is that the interaction between these values is considered and one can compensate for the other, ensuring that the most significant probes are selected. Subsequently, we performed a receiver’s operating characteristic (ROC) curve analysis and retained 500 probes with the highest area under the ROC curve (AUC). Finally, we eliminated probes with a pair-wise correlation greater than 0.95 measured using Pearson’s correlation coefficients for all probes, for the case and control samples separately. We then performed hierarchical clustering using the remaining 175 probes, by Ward’s method on Euclidean distance using the gplots package. More details about the 175 selected probes are summarized in Appendix A. The methylation levels (β values) at those probes for Patients 1–6 and for the control samples have also been provided in Appendix A. Multidimensional scaling (MDS) was done by scaling of the pair-wise Euclidean distances between samples. We constructed a binary support vector machine (SVM) using the e1071 package as described previously [4,8,19]. In order to detect the differentially methylated regions (DMRs), we used the DMRcate package [26], and regions containing at least 5 different CpGs within 1kb with a minimum methylation difference of 10% between the case and control groups and a Fisher’s multiple comparison *p*-value < 0.01 were selected.

## 3. Results

### 3.1. Detection and Verification of an Episignature for MRXSA

The case samples included 6 males from three different families (Patients 1–4 from one family, and Patients 5 and 6 from the two other families) (Table 1). We had two samples from Patient 4 collected at ages 4 and 28 years. We used the sample from 28 years old (sample A) for probe selection and training unsupervised and supervised models, and the sample from 4 years old (sample B) as a testing sample. Patients 1–4 have the same *FAM50A* variant, c.764A>G; p.Asp255Gly, and Patients 5 and 6 have variants c.761A>G; p.Glu254Gly and c.763G>A; p.Asp255Asn, respectively. All the variants were classified as pathogenic or likely pathogenic, according to the American College of Medical Genetics (ACMG) guidelines. Variants of all patients, except for Patients 5 and 6, were inherited [21]. 

Patient 1 last underwent a clinical examination at the age 62 and was institutionalized since he was 24. He had seizures in infancy, presented with global developmental delay (GDD), and started walking at age 7. He developed bilateral open-angle glaucoma and bilateral cataracts later in adulthood. He also had short stature, speech problems, and craniofacial anomalies [20]. The clinical presentations of patient 2 included short stature, dysmorphic facial features, and a left inguinal hernia [20]. Patient 3 manifested GDD, speech problems, seizures, short stature, craniofacial anomalies, glaucoma, and small hands and feet [20]. Patient 4 had clinical features including GDD, dysmorphic facial features, strabismus, and small feet [21]. More detailed clinical information of this patient at different ages can be found in [20,21]. Patient 5 presented with GDD, dysmorphic facial features, strabismus, and short stature [21], while patient 6 presented with GDD, dysmorphic facial features, and exotropia [21]. 

We selected 42 control samples from our database matched for age, sex, and array type (EPIC) (using the case to control ratio of 7). The 175 probes selected using the three-step process described in the Methods section were used for the purpose of constructing unsupervised and supervised classification models. The methylation levels at these 175 CpG sites are considered as the identifying episignature of the syndrome.

In order to assess the robustness of the episignature in differentiating between the case and control samples, we performed hierarchical clustering (Figure 1A) observing a clear separation of the two groups. We observed similar separation using multidimensional scaling (MDS) analysis (Figure 1B). We then re-conducted the hierarchical clustering and MDS models with the initial 6 case samples and 42 control samples as the training set, after including one other control sample and sample B from Patient 4 as the testing set. As expected, in both plots the testing control sample and sample B from Patient 4 were correctly clustered with their corresponding classes (Figure 2). An interesting observation is that sample B from Patient 4 demonstrates a slightly different methylation profile from the other case subjects. This can be due to the fact that these samples were extracted from the patient at different ages (24 years apart).

We also performed 6-fold cross-validation multidimensional scaling, selecting probes using 5 case samples as the training set and 1 case sample as the testing set at each step. In all steps, the testing sample was correctly clustered with the training case samples, further providing evidence of a robust common DNA methylation signature (Figure 3).

3.2 Construction of the Binary Prediction Model 

For the purpose of classifying case and control samples more accurately, we constructed a binary SVM classifier with a linear kernel, using the selected probes (see the details in [19]). For each sample, the classifier creates a methylation variant probability (MVP) score between 0 and 1. A sample is identified as having a methylation pattern similar to the signature detected for the syndrome if the MVP score is near 1, and it is indicated as having a methylation behavior similar to controls otherwise. 

First, we only used samples from controls and samples from individuals with MRXSA for training the model, and supplied over 1000 control subjects and cases of 38 other constitutional disorders with episignatures from the EKD [19] (Table 2) into the model in order to assess the specificity of our classifier. The classifier showed a high sensitivity for MRXSA, with all samples scoring high on the MVP axis (Figure 4). The specificity, defined as the MVP score >0.5 was over 99%, with 5 samples from other disease cohorts and controls (totaling >1000 samples) scoring above that cut-off. Three of those cases are from the cohort of patients with Chr5q35-qter duplication and a clinical diagnosis of Hunter McAlpine craniosynostosis syndrome, suggesting level of similarity in the corresponding episignature.

In order to increase the accuracy of the classifier, we retrained our model using all the MRXSA subjects, 75% of healthy control subjects, and 75% of patients from 38 other neurodevelopmental disorders [19] (listed in Table 2) as the training set and the remaining 25% as the testing set. We should mention that because of the small case sample size, all the MRXSA samples were used for model training. This step allows the preferential selection of probes that are not overlapping with other genetic disorders and improve the specificity of the classifier. This improved the classifier and allowed us to differentiate between the testing samples from the MRXSA cohort and the remaining disease and reference cohorts. It confirmed the existence of an MRXSA episignature shared between the 6 subjects and 3 families. As expected, sample B from Patient 4 received a lower, yet distinctly elevated score compared to the rest of the control and other disorder samples (Figure 5). 

### 3.3. Identification of the Regions of Differential Methylation

We used the detected MRXSA episignature to search for DMRs. We identified 55 regions of differential methylation (Appendix A). A region on chromosome 22 with the largest number of significant CpGs overlapped *CPT1B*, which has a role in cardiac development [35], and *CHKB*, with a function in the formation of skeletal muscles [36]. 

## 4. Discussion

Over 50 genes and more than 42 neurodevelopmental conditions are currently described with associated DNA methylation signatures, also referred to as episignatures or EpiSigns [19,37]. Many of the related genes have a regulatory role in the epigenetic machinery; such as histone modification, DNA methylation, or chromatin remodeling. We and others have shown the diagnostic utility of genome-wide DNA methylation analysis using peripheral blood [9,13,19,34,38]. This method has been applied to assign a diagnosis to many of the ND/CA-affected subjects that remained unresolved by conventional testing [4,8,9,17,18,19,32,33,34]. This approach has also been effective in deriving a correct genetic diagnosis in patients with incorrect initial clinical diagnosis [4]. More recently, the test called EpiSign, was adapted as the first genome-wide DNA methylation clinical test for patients with ND/CA which can be used either as part of diagnostic assessment or for reclassification of previously detected VUSs (https://genomediagnostics.amsterdamumc.nl/epigenetic-test/; https://www.ggc.org/episign).

The intellectual developmental disorder, X-linked, syndromic, Armfield type (MRXSA) was described in 1999, as an intellectual disability disorder that presents with features including global developmental delay, short stature, seizure, craniofacial anomalies, and ocular abnormalities such as glaucoma [20]. Missense variants in *FAM50A* were recently reported as the causal variants of the disorder [21]. Here, based on DNA methylation samples collected from peripheral blood of 6 patients from 3 families, we identified an episignature specific to the syndrome. 

DNA methylation episignatures vary in their genomic locations and the robustness or the DNA methylation difference across different disorders. MRXSA episignature is of the more robust type, enabling a discovery and validation of the highly sensitive and specific signal in a relatively small number of patient samples. More detailed information about the 175 selected probes are summarized in Appendix A. In particular, the gene they are located on has been indicated, some of them having a regulatory role in development (for instance, *CPT1B*). Figure 4 illustrates full sensitivity and specificity of our model, where all case samples received a very high MVP score and all control samples and individuals from the other 38 constitutional disorders received a score near zero. Notably, sample B from Patient 4 received a low MVP score compared to the rest of MRXSA samples. This is probably because the blood samples were extracted from the patient 24 years apart, and the methylation profile has changed during this time. Alternatively, some of the contributing factors may be related to the specimen quality and storage, and wet-lab sample processing effects. While it is an established fact that DNA methylation patterns are amongst the most accurate biomarkers of the aging process, this finding is also in line with our previous observations that a loss-of-function mutations in NSD1, which causes another EpiSign disorder, Sotos syndrome, substantially accelerates epigenetic aging [39]. One limitation of genome-wide methylation analysis for Mendelian neurodevelopmental disorders is that these syndromes are generally very rare. We expect that by increasing the number of samples and expanding the range and type of variants we may uncover further sub-stratification of the DNA methylation profile of *FAM50A*, as we have previously observed for conditions including Weaver syndrome (WVS) and Coffin–Siris syndrome (CSS) [8,17,40]. 

In addition to a distinct genome wide episignature that can be used as a sensitive diagnostic biomarker, we also identified 55 regions of differential methylation in patients with *FAM50A* variants. The most significant region contained 17 CpG sites, overlapping two genes on chromosome 22 (*CHKB* and *CPT1B*) with a role in pattern formation and development. Other genes with functions in regulation of developmental processes included *LIMS3*, which has a role in neural tissue patterning and differentiation [41]; *PRDM9*, which is involved in histone modification and hence, in regulating the epigenetic machinery [42]; and *CACNA1C*, which is associated with Timothy syndrome (TS), a rare neurodevelopmental disorder [43]. While methylation changes in these regions point to the possibility of the associated pathophysiology, further functional and integrative genomics analysis would be necessary to study the possible causation or correlation. Also, while the use of peripheral blood for discovery of episignatures makes these findings broadly applicable to routine diagnostic testing of patients with rare disorders, to better understand the pathophysiology of these epigenetic changes, it will be important to study other tissues most significantly affected by the clinical symptoms, including neuronal tissues.

In conclusion, the discovery of the MRXSA DNA methylation episignature adds to the list of Mendelian neurodevelopmental disorders with DNA methylation episignatures that can be used for screening and diagnosis of patients with rare neurodevelopmental conditions. Additional work focused on expanding the number of cases and variant types across the *FAM50A* gene is necessary to further refine this episignature and assess possible additional DNA methylation profiles associated with this disorder. 

## Figures and Tables

**Figure 1 ijms-22-01111-f001:**
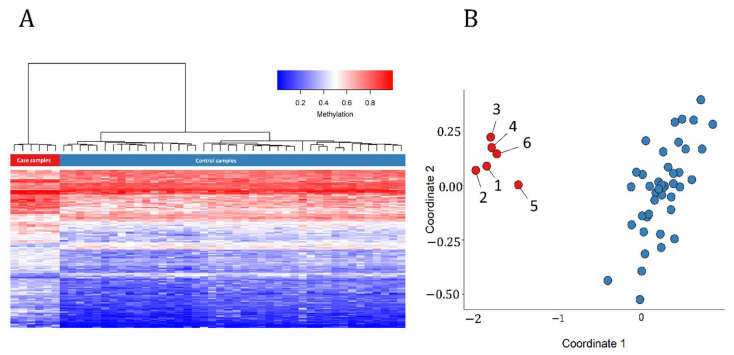
Assessment of the strength of the identified episignature in distinguishing intellectual developmental disorder, X-linked, syndromic, Armfield type (MRXSA) case subjects from controls. (**A**) Hierarchical clustering with Ward’s method on Euclidean distance was performed. In the heatmap, each row illustrates a selected CpG site, and each column is related to a sample. The heatmap pane represents the phenotype. The heatmap color scale indicates the range of methylation level; from blue (no methylation or 0) to red (full methylation or 1). This plot conveys that the detected episignature clearly differentiates between case and control subjects. (**B**) Multidimensional scaling plot using the selected probes, illustrating the power of the signature in separating the case and control groups. Blue circles represent healthy control subjects and red circles indicate subjects with a confirmed diagnosis of the syndrome.

**Figure 2 ijms-22-01111-f002:**
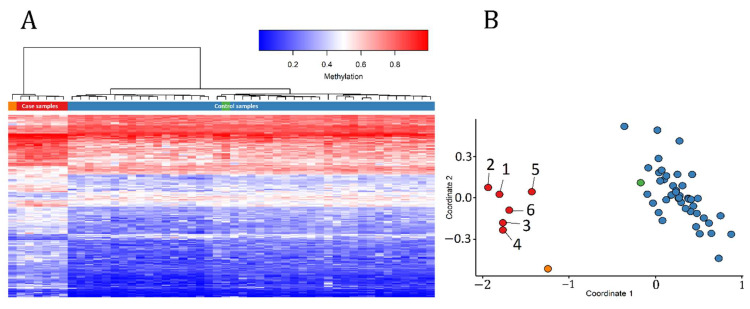
Adding testing samples to our unsupervised models. Using the case and control samples as the training set and one other control and sample B from Patient 4 as the testing set. Green represents the testing control sample and orange represents sample B from Patient 4. (**A**) Hierarchical clustering, (**B**) multidimensional scaling. It is observed that sample B from Patient 4 demonstrates a slightly different methylation pattern from the rest of the case samples and it clusters farther from the case group on the MDS plot.

**Figure 3 ijms-22-01111-f003:**
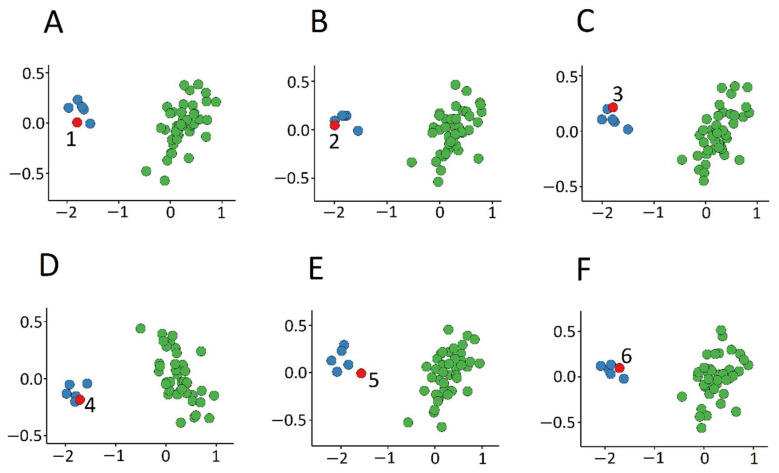
6-fold cross-validation multidimensional scaling. In each of the steps (**A**–**F**), the red circle represents one of the Patients 1–6, respectively.

**Figure 4 ijms-22-01111-f004:**
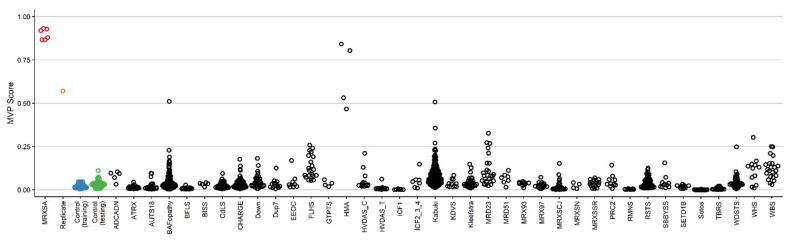
The methylation variant probability (MVP) scores created by the support vector machine (SVM) trained by comparing the 6 cases of MRXSA against healthy control subjects. The red circles represent the case samples, the orange circle represents sample B from Patient 4, the black circles represent cases from the other 36 neurodevelopmental disorders and congenital anomalies (ND/CAs), the blue circles represent training control samples, and the green circles represent testing control samples.

**Figure 5 ijms-22-01111-f005:**
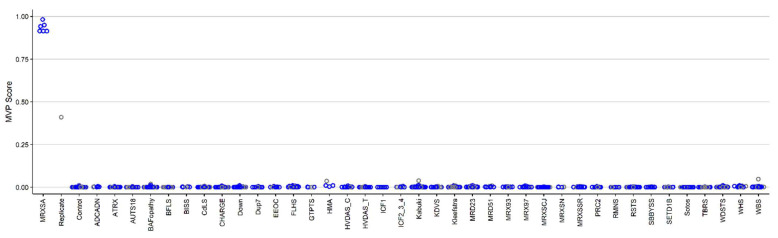
The MVP scores created by the SVM trained by comparing cases of MRXSA against healthy control subjects and samples from 38 other constitutional EpiSign disorders. The blue circles represent the training samples and the grey circles the testing samples.

**Table 1 ijms-22-01111-t001:** Clinical and genetic information of the patients.

Patient	Kindred	Age	*FAM50A* Variant	Clinical Features
1	8100	62	c.764A>G; p.Asp255Gly inherited	Global developmental delay (GDD), glaucoma, cataracts, short stature, speech problems, and craniofacial anomalies
2	8100	50	c.764A>G; p.Asp255Gly inherited	Short stature, dysmorphic facial features, and a left inguinal hernia
3	8100	45	c.764A>G; p.Asp255Gly inherited	GDD, speech problems, seizures, short stature, craniofacial anomalies, glaucoma, and small hands and feet
4	8100	28	c.764A>G; p.Asp255Gly inherited	GDD, dysmorphic facial features, strabismus, and small feet
5	9656	10	c.761A>G; p.Glu254Gly de novo	GDD, strabismus, short stature, and dysmorphic facial features
6	9677	26	c.763G>A; p.Asp255Asn de novo	GDD, dysmorphic facial features, and exotropia

**Table 2 ijms-22-01111-t002:** List of syndromes with a defined episignature, used for training a more accurate SVM.

Syndrome	Syndrome Abbreviation	Underlying Gene/Location	Phenotype MIM Number	Signature Published
Cerebellar ataxia, deafness, and narcolepsy, autosomal dominant	ADCADN	*DNMT1*	604121	Yes [4,8,19,27]
Alpha-thalassemia mental retardation syndrome	ATRX	*ATRX*	301040	Yes [4,7,8,19]
Autism, susceptibility to, 18	AUTS18	*CHD8*	615032	Yes [19,28]
BAFopathies: Coffin–Siris 1–4 (CSS1–4) and Nicolaides-Baraitser (NCBRS) syndromes	BAFopathy	*ARID1A*, *ARID1B*, *SMARCB1*, *SMARCA4*, *SMARCA2*	614607, 135900, 614609, 614608, 601358	Yes [4,17,19]
Börjeson-Forssman-Lehmann syndrome	BFLS	*PHF6*	301900	Yes [19]
Blepharophimosis intellectual disability syndrome	BIS	*SMARCA2*	NA	Yes [29]
Cornelia de Lange syndrome 1–4	CdLS	*NIPBL*, *RAD21*, *SMC3*, *SMC1A*	122470, 614701, 610759, 300590	Yes [4,19]
CHARGE syndrome	CHARGE	*CHD7*	214800	Yes [4,8,9,19]
Down syndrome	Down	Chr21 trisomy	190685	Yes [4,19,30]
Chr7q11.23 duplication syndrome	Dup7	Chr7q11.23 Duplication	609757	Yes [4,19,31]
Epileptic encephalopathy, childhood-onset	EEOC	*CHD2*	615369	Yes [19]
Floating-Harbor syndrome	FLHS	*SRCAP*	136140	Yes [4,8,15,19]
Genitopatellar syndrome	GTPTS	*KAT6B*	606170	Yes [4,8,19]
Hunter McAlpine craniosynostosis syndrome	HMA	Chr5q35-qter duplication involving NSD1	601379	Yes [19]
Helsmoortel-van der Aa syndrome (ADNP syndrome [Central])	HVDAS_C	ADNP (c.2000-2340)	615873	Yes [4,19]
Helsmoortel-van der Aa syndrome (ADNP syndrome [Terminal])	HVDAS_T	ADNP (outside c.2000-2340)	615873	Yes [4,19]
Immunodeficiency-centromeric instability-facial anomalies syndrome 1	ICF1	*DNMT3B*	242860	Yes [19]
Immunodeficiency-centromeric instability-facial anomalies syndrome 2–4	ICF2-4	*CDCA7*, *ZBTB24*, *HELLS*	614069, 616910, 616911	Yes [19]
Kabuki syndrome 1 and 2	Kabuki	*KMT2D*, *KDM6A*	147920, 300867	Yes [4,8,9,19,32]
Koolen de Vries syndrome	KDVS	*KANSL1*	610443	Yes [19]
Kleefstra syndrome 1	Kleefstra1	*EHMT1*	610253	Yes [19]
Mental retardation, autosomal dominant 23	MRD23	*SETD5*	615761	No
Mental retardation, autosomal dominant 51	MRD51	*KMT5B*	617788	Yes [19]
Mental retardation, X-linked 93	MRX93	*BRWD3*	300659	Yes [19]
Mental retardation, X-linked 97	MRX97	ZNF711	300803	Yes [19]
Mental retardation, X-linked, syndromic, Claes-Jensen type	MRXSCJ	*KDM5C*	300534	Yes [4,8,11,19]
Mental retardation, X-linked syndromic, Nascimento-type	MRXSN	*UBE2A*	300860	Yes [19]
Mental retardation, X-linked, Snyder- Robinson type	MRXSSR	*SMS*	309583	Yes [19]
PRC2: Cohen-Gibson syndrome (COGIS) and Weaver syndrome (WVS)	PRC2	*EED*, *EZH2*	617561, 277590	No
Rahman syndrome	RMNS	*HIST1H1E*	617537	Yes [19,33]
Rubinstein-Taybi syndrome 1 and 2	RSTS	*CREBBP*, *EP300*	180849, 613684	Yes [19]
Ohdo syndrome, SBBYS variant	SBBYSS	*KAT6B*	603736	Yes [8,19]
SETD1B-related syndrome	SETD1B	*SETD1B*	N/A	Yes [34]
Sotos syndrome	Sotos	*NSD1*	117550	Yes [4,8,13,19]
Tatton-Brown-Rahman syndrome	TBRS	*DNMT3A*	615879	Yes [19]
Wiedemann-Steiner syndrome	WDSTS	*KMT2A*	605130	Yes [19]
Williams-Beuren syndrome	WBS	Chr7q11.23 deletion	194050	Yes [4,19,31]
Wolf-Hirschhorn syndrome	WHS	Chr4p16.3 deletion	194190	No

## Data Availability

Some of the datasets used in this study are publicly available and may be obtained from gene expression omnibus (GEO) using the following accession numbers. GEO: GSE116992, GSE66552, GSE74432, GSE97362, GSE116300, GSE95040, GSE 104451, GSE125367, GSE55491, GSE108423, GSE116300, GSE 89353, GSE52588, GSE42861, GSE85210, GSE87571, GSE87648, GSE99863, and GSE35069. These include DNA methylation data from patients with Kabuki syndrome, Sotos syndrome, CHARGE syndrome, immunodeficiency-centromeric instability-facial anomalies (ICF) syndrome, Williams-Beuren syndrome, Chr7q11.23 duplication syndrome, BAFopathies, Down syndrome, a large cohort of unresolved subjects with developmental delays and congenital abnormalities, and also several large cohorts of DNA methylation data from the general population. The rest of the data including the MRXSA samples are not available due to the restrictions of the ethics approval.

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
