# Peer review of "Detection of a DNA Methylation Signature for the Intellectual Developmental Disorder, X-Linked, Syndromic, Armfield Type"

_ijms, 2021, doi:10.3390/ijms22031111_

Round 1

Reviewer 1 Report

The authors of this paper describe epimutations associated with a genetic developmental disorder, and derive a model to provide a diagnosis of this disorder with a clear clinical application. 

However, the description of why this is important/the scientific significance as well as an adequate description of the methods is lacking. 

Specifically: 

  1. In the introduction there is no discussion of how common or rare these disorders are. Adding such a discussion would help readers unfamiliar with neurological disorders understand why this is important problem. 
  2. I struggled to understand the significance of the epimutation vs. the genetic mutations. It's unclear what additional information the epimutations give, other than being overlapping between different genetic profiles. 
  3. In the methods section a lot more description is required, specifically: (a) a better description of the patients and samples, specifically the patient who is sampled twice should be included at the beginning (b) exclusion criteria for probes: why are x, y and SNP probes being removed (c) the rational for selection of 1000 probes to investigate further is warranted as opposed to using a p-value or other statistical measure. 
  4. Age is never addressed by a covariate at any point in the analysis despite epigenetic signatures changing over time. An overlay of the epigenetic clock sites and the signature sites would be helpful to understand if these positions are impacted by age. 
  5. A discussion of power or other contributing factors in the determination of the test vs. training set would be useful. It is unclear in the current version of the manuscript whether the 6 to 1 design is statistically sufficient. 

Author Response

Reviewer reports:

Reviewer #1: The authors of this paper describe epimutations associated with a genetic developmental disorder, and derive a model to provide a diagnosis of this disorder with a clear clinical application. 

However, the description of why this is important/the scientific significance as well as an adequate description of the methods is lacking. 

Specifically: 

1) In the introduction there is no discussion of how common or rare these disorders are. Adding such a discussion would help readers unfamiliar with neurological disorders understand why this is important problem. 

Response: Thank you. We have added a sentence and a reference to the introduction that provides the prevalence of rare genetic diseases and emphasizes the importance of a correct diagnosis for the affected individuals.

“The frequency of Mendelian disorders is approximated to be 40 to 82 per 1,000 live births (Christianson A, Howson C, Modell B. March of Dimes. Global report on birth defect. The hidden toll of dying and disabled children. New York. Published online 2006). Considering all congenital anomalies, 8% of individuals are estimated to have a genetic disorder before adulthood (Baird, P.A.; Anderson, T.W.; Newcombe, H.B.; Lowry, R.B. Genetic disorders in children and young adults: A population study. Am. J. Hum. Genet. 1988, 42, 677–693). Given the broad range of genetic and phenotypic heterogeneity, based on an individual’s presentation and clinical assessment alone, it is often impossible to determine the precise clinical diagnosis in the absence of a specific molecular genetic diagnosis.”

2) I struggled to understand the significance of the epimutation vs. the genetic mutations. It's unclear what additional information the epimutations give, other than being overlapping between different genetic profiles. 

Response: We use a term episignature throughout the manuscript. Episignature is related to the term genetic mutation in that it represents a consequence, or DNA methylation profile, resulting from the underlaying DNA mutation. We have expanded paragraph 2 of the introduction to further clarify this point:

Recent advances in epigenetic analysis have provided an alternate approach for diagnosis of genetic disorders. Pathogenic variants in genes that encode proteins involved in the epigenetic machinery, chromatin assembly and transcription regulation can cause changes in the genome-wide pattern of DNA methylation, differentiating them from unaffected individuals. These highly sensitive and specific changes in DNA methylation patterns, referred to as episignatures, are currently used to help reclassify VUS’s as likely pathogenic or benign, thus enabling a definitive diagnosis [5]. Hence, a term episignature is used to describe a consequence of a unique DNA methylation pattern, resulting from the underlaying DNA mutation.”

3) In the methods section a lot more description is required, specifically: (a) a better description of the patients and samples, specifically the patient who is sampled twice should be included at the beginning (b) exclusion criteria for probes: why are x, y and SNP probes being removed (c) the rational for selection of 1000 probes to investigate further is warranted as opposed to using a p-value or other statistical measure. 

Response: a) a description of the replicated sample has been added to the first paragraph of the Materials and Methods section. b) An explanation of the rationale behind the removal of such probes has been added. c) The advantage of this approach over strict cut-off values is now provided.

4) Age is never addressed by a covariate at any point in the analysis despite epigenetic signatures changing over time. An overlay of the epigenetic clock sites and the signature sites would be helpful to understand if these positions are impacted by age. 

Response: In the methods section 2.2 that describes the probe selection procedure we state that “Using the MatchIt R package [22], for each case we selected seven controls matched for age, sex, and array type from the EpiSign Knowledge Database (EKD)”. Hence the analysis is normalized for probes/regions that may exhibit age-related changes in DNA methylation.

5) A discussion of power or other contributing factors in the determination of the test vs. training set would be useful. It is unclear in the current version of the manuscript whether the 6 to 1 design is statistically sufficient. 

Response: An explanation has been added to the end of Section 2.2, providing the reason for the choice of the ratio 7.

Reviewer 2 Report

The present manuscript by Haghshena et al., entitled:

‘Detection of a DNA Methylation Signature for the Intellectual Developmental Disorder, X-linked, Syndromic, Armfield Type’

reports the distinct DNA methylation episignature observed in 6 individuals with Developmental Disorder, X-linked, Syndromic, Armfield Type (MRXSA), a disorder caused by missense variants in FAM50A. Analysis were done in peripheral blood of 6 patients aged 10 to 62 (four of them members of the same family and with the same variant in FAM50A; and two other individuals with different variants in FAM50A), using Illumina Infinium methylation EPIC bead chip arrays that includes over 850,000 CpG sites. After applying a 3-step biostatistical pipeline to identify the most differing epigenetic sites compared with matched controls, authors selected 175 CpG sites and considered them as the identifying episignature of the syndrome. Additionally, they also identified 55 regions of differential methylation that overlapped different genes that they detailed. Authors conclude that the detected episignature clearly differentiates between case and control subjects.

With the information on those 175 probes, authors developed a binary prediction model that they trained with the episignature of the MRXSA patients, as well as of patients from 38 other constitutional EpiSign disorders and controls. The model was able to identify individuals with MRXSA and suggested similarity with the episignature of Hunter McAlpine craniosynostosis syndrome and Chr5q35-qter duplication. The authors conclude that the provided data can be used for the diagnosis of MRXSA.

Despite the number of samples used in this study is low (only 6 patients), the episignature of MRXSA patients seems robust and offers significant information that could additionally be useful for future studies, when more samples of patients with this low incidence disorder are available. Thus, I recommend this study for publication.

Some minor comments follow:  

  • Since in the present study the number of patients is low, all results that can be useful for future studies with higher number of patients should be included as supplemental material (Eg. Information on the 175 CpG sites considered as the identifying episignature of the syndrome, DMRS, details and script of the binary prediction model, etc.).
  • M&M to detect the differentially methylated regions (DMRs), was it done one by one individual? Please specify.
  • Table S1: Would it be possible to include the results of the analysis for each individual? (E.g. one different sheet of the excel for each patient).
  • The causal locus of MRXSA is localized in Xq28. Was differential methylation observed in this region? Are the arrays detecting in this region? And in FAM50A?
  • Considering that MRXSA is caused by missense variants in FAM50A, is it common that regions of differential methylation do not overlap the causal gene of the disorder, as it occurs here with FAM50A? Please, discuss and provide references.
  • Considering that about 70% of promoters located near the transcription start site of a gene (proximal promoters) contain a CpG island, would it be possible to assess if those selected 175 CpG sites are in promoters? And if so, in the promoters of what genes?
  • Authors observed differences in Sample A and B of patient 4 which were sampled at different ages of the same individual (4 and 28 years old): could they be due to a problem in storage and conservation of the sample? The range of age among the other patients was even higher (10-62 y.o.), but still these differences could not be observed… What other reasons could you propose to explain differences in Sample A and B? Please discuss accordingly.
  • Discussion: Please, briefly discuss on the 175 CpG selected sites.

Misspelling:

Fig 1: panel (instead of pane).

Discussion: Please, include a comma before Sotos syndrome.

Author Response

Reviewer #2: The present manuscript by Haghshena et al., entitled:

‘Detection of a DNA Methylation Signature for the Intellectual Developmental Disorder, X-linked, Syndromic, Armfield Type’

reports the distinct DNA methylation episignature observed in 6 individuals with Developmental Disorder, X-linked, Syndromic, Armfield Type (MRXSA), a disorder caused by missense variants in FAM50A. Analysis were done in peripheral blood of 6 patients aged 10 to 62 (four of them members of the same family and with the same variant in FAM50A; and two other individuals with different variants in FAM50A), using Illumina Infinium methylation EPIC bead chip arrays that includes over 850,000 CpG sites. After applying a 3-step biostatistical pipeline to identify the most differing epigenetic sites compared with matched controls, authors selected 175 CpG sites and considered them as the identifying episignature of the syndrome. Additionally, they also identified 55 regions of differential methylation that overlapped different genes that they detailed. Authors conclude that the detected episignature clearly differentiates between case and control subjects.

With the information on those 175 probes, authors developed a binary prediction model that they trained with the episignature of the MRXSA patients, as well as of patients from 38 other constitutional EpiSign disorders and controls. The model was able to identify individuals with MRXSA and suggested similarity with the episignature of Hunter McAlpine craniosynostosis syndrome and Chr5q35-qter duplication. The authors conclude that the provided data can be used for the diagnosis of MRXSA.

Despite the number of samples used in this study is low (only 6 patients), the episignature of MRXSA patients seems robust and offers significant information that could additionally be useful for future studies, when more samples of patients with this low incidence disorder are available. Thus, I recommend this study for publication.

Some minor comments follow:  

1) Since in the present study the number of patients is low, all results that can be useful for future studies with higher number of patients should be included as supplemental material (Eg. Information on the 175 CpG sites considered as the identifying episignature of the syndrome, DMRS, details and script of the binary prediction model, etc.).

Response: The 175 CpG sites have been attached to the submission as an excel file (Table S2). The DMRs are also included in Table S1, uploaded as an excel file. Also, the methylation levels at the 175 selected CpG sites for the case and control samples have been provided in Table S3. A detailed, stepwise, analytical procedure has been described in the cited papers (PMID: 32109418; PMID: 30929737).

2) M&M to detect the differentially methylated regions (DMRs), was it done one by one individual? Please specify.

Response: The phrase “between the case and control groups” has been added, indicating that the analysis has not been performed on an individual basis.

3) Table S1: Would it be possible to include the results of the analysis for each individual? (E.g. one different sheet of the excel for each patient).

Response: The data analysis presented in this paper has been done as a comparison between the two groups rather than individually to increase the statistical power. We have provided the related information summarized in Table S1.

4) The causal locus of MRXSA is localized in Xq28. Was differential methylation observed in this region? Are the arrays detecting in this region? And in FAM50A?

Response: According to Table S1, the DMRs did not overlap the region of Xq28 or FAM50A.

5) Considering that MRXSA is caused by missense variants in FAM50A, is it common that regions of differential methylation do not overlap the causal gene of the disorder, as it occurs here with FAM50A? Please, discuss and provide references.

Response: Yes, this is a typical result in genome-wide DNA methylation analysis. We have added the following to the introduction to further elaborate this point:

“These genome wide episignatures are the consequence of genetic mutations resulting in a defective function of the related protein. Regions with significant disruptions in DNA methylation can range from hundreds to tens of thousands of probes in the methylation array, but can show partial overlap between different disorders, and normally do not involve disruption of DNA methylation in the related gene (PMID: 32109418)”

6) Considering that about 70% of promoters located near the transcription start site of a gene (proximal promoters) contain a CpG island, would it be possible to assess if those selected 175 CpG sites are in promoters? And if so, in the promoters of what genes?

Response: A table (Table S2) has been added to the submission with information on the 175 CpG sites. The columns “In the promoter?” and “UCSC gene name(s)” specify the probe location.

7) Authors observed differences in Sample A and B of patient 4 which were sampled at different ages of the same individual (4 and 28 years old): could they be due to a problem in storage and conservation of the sample? The range of age among the other patients was even higher (10-62 y.o.), but still these differences could not be observed… What other reasons could you propose to explain differences in Sample A and B? Please discuss accordingly.

Response:  We agree with the reviewer. It is uncertain what is the cause of the relative change in the DNA methylation profile. While we proposed individual age-related changes as a possibility, technical/sample related differences may be contributory. We have added further comment:

Notably, sample B from Patient 4 received a low MVP score compared to the rest of MRXSA samples. This is probably because the blood samples were extracted from the patient 24 years apart, and the methylation profile has changed during this time. Alternatively, some of the contributing factors may be related to the specimen quality and storage, and wet-lab sample processing effects.

8) Discussion: Please, briefly discuss on the 175 CpG selected sites.

Response: A brief discussion has been included in the Discussion section.

Misspelling:

9) Fig 1: panel (instead of pane).

Response: The word “pane” (which should have been “panel”) was removed due to redundancy.

10) Discussion: Please, include a comma before Sotos syndrome.

Response:  Thank you. We have corrected the topographical error.

Round 2

Reviewer 1 Report

The authors have addressed all my concerns and I have no further considerations.